# A Polytetrafluoroethylene-Based Solvent-Free Procedure for the Manufacturing of Lithium-Ion Batteries

**DOI:** 10.3390/ma16227232

**Published:** 2023-11-19

**Authors:** Xuehan Wang, Shuli Chen, Kaiqi Zhang, Licheng Huang, Huilin Shen, Zheng Chen, Changru Rong, Guibin Wang, Zhenhua Jiang

**Affiliations:** 1Key Laboratory of High-Performance Plastics, Ministry of Education, National and Local Joint Engineering Laboratory for Synthesis Technology of High-Performance Polymers, College of Chemistry, Jilin University, Changchun 130012, China; xuehan23@mails.jlu.edu.cn (X.W.); zhangkq22@mails.jlu.edu.cn (K.Z.); shenhl22@mails.jlu.edu.cn (H.S.); wgb@jlu.edu.cn (G.W.); jiangzhenhua@jlu.edu.cn (Z.J.); 2National Key Laboratory of Advanced Vehicle Integration and Control, China FAW Group Co., Ltd., Changchun 130013, China; chenshuli@faw.com.cn (S.C.); huanglicheng@faw.com.cn (L.H.)

**Keywords:** lithium-ion battery (LIBs), polymer binder, solvent-free (SF) procedure, polytetrafluoroethylene (PTFE)

## Abstract

Lithium-ion batteries (LIBs) have recently become popular for energy storage due to their high energy density, storage capacity, and long-term cycle life. Although binders make up only a small proportion of LIBs, they have become the key to promoting the transformation of the battery preparation process. Along with the development of binders, the battery manufacturing process has evolved from the conventional slurry-casting (SC) process to a more attractive solvent-free (SF) method. Compared with traditional LIBs manufacturing method, the SF method could dramatically reduce and increase the energy density due to the reduced preparation steps and enhanced electrode loading. Polytetrafluoroethylene (PTFE), as a typical binder, has played an important role in fabricating high-performance LIBs, particularly in regards to the SF technique. In this paper, the development history and application status of PTFE binder was introduced, and then its contributions and the inherent problems involved in the SF process were described and analyzed. Finally, the viewpoints concerning the future trends for PTFE-based SF manufacturing methods were also discussed. We hope this work can inspire future research concerning high-quality SF binders and assist in promoting the evolution of the SF manufacturing technology in regards to LIBs.

## 1. Introduction

With the increasing global energy consumption, it is urgent to investigate alternative low-carbon and ecologically friendly energy sources to minimize reliance on fossil fuels and meet the rising demand for energy storage [1,2,3,4]. Lithium-ion batteries (LIBs) [4,5] are one of the most promising energy technologies. They are rapidly gaining popularity in new energy vehicles, intelligent gadgets, and electronic devices due to their high energy density, excellent efficiency, and long cycle life [6,7,8]. However, while LIBs present opportunities, they also reveal significant inherent challenges. Previous work [9,10] has shown that the electrode preparation process significantly affects the performance and capability of LIBs. Their manufacturing costs and the pollution emitted by recycling waste batteries require further reduction. 

Currently, the slurry-casting (SC) process is used to prepare most commercialized LIBs electrodes [11,12]. First, the active material, conductive agent, and binder are homogeneously mixed under solvent-free conditions; then, deionized water [12] or N-methyl pyrrolidone (NMP) [13,14], etc., solvent is added to prepare a slurry with suitable viscosity for casting on the electrode collector; after that, the slurry is coated on the electrode collector; and finally, the finished electrodes are produced by heating, rolling, and other processes. However, there are several problems with the SC process. (1) Energy consumption for solvent evaporation. Solvents such as NMP have to be dried using large production lines for rapid evaporation from SC electrodes. The total energy consumption is 51% of the production line using 1 million batteries (20.5 Ah, 3.7 V) per year [15]. Moreover, Chris Yuan et.al found that drying energy consumption accounts for 38% of a production line, with a total of 88.9 GJ of primary energy consumed in producing the 24 kWh LMO-graphite battery pack, as shown in Figure 1a [16]. (2) Battery recycling difficulties. Expensive and complex equipment is often required to recover NMP, which also increases the budget. In 2016, Shabbir Ahmed et al. [17] presented a model for studying the cost of drying the electrodes and recycling the NMPs in a LIBs plant. The cost of the NMP recovery steps amount to USD 4.6 million per year, or USD 45.8 per pack, as shown in Figure 1b. (3) Uneven electrode materials. The faster the solvent evaporates, the more prone it is to gradient changes in binder concentration at various locations in the paste, leading to rapid failure of the batteries [18,19,20]. (4) Biological toxicity. NMP reveals a direct embryotoxic potential [21,22].

In order to decrease the production costs and environmental pollution resulting from electrode fabrication, researchers have been continuously exploring the development of a more efficient electrode preparation process. Wood et al. proposed two strategies: (1) reducing the costs associated with organic solvents and drying time, and (2) increasing the electrode thickness, without affecting power density. The above methods could significantly reduce the costs of LIB [23], resulting in an enhanced capability for sustainable production within the renewable energy industry. In other words, without binder or solvent, the electrode preparation process can be drastically simplified, and both the cost and environmental pollution will diminish sharply. Current research work regarding binder-free LIBs [24,25] is still at the laboratory level, and the performance of these LIBs remains far short of meeting commercialization requirements. However, there have been studies [26,27] showing that solvent-free (SF) electrode preparation techniques can enable commercialized production. For example, the method of polymer fibrillation [28,29,30], with low cost and mass production capability, has been considered as one of the mainstream methods for the preparation of dry electrode membranes. The binders commonly used for polymer fibrillation include polytetrafluoroethylene (PTFE), copolymers of PTFE with other monomers (e.g., ethylene, hexafluoropropylene), etc. [31]. Among them, PTFE is considered to be the most suitable adhesive for the following reasons: (1) it has larger polymerized molecular weight; (2) it exhibits high inertness and corrosion resistance; and (3) it possesses good mechanical properties [32].

Up until now, SF technology, particularly the polymer fibrillation method, has been mainly developed by and reserved for industry. Although there is currently no doubt regarding the place of PTFE in the polymer fibrillation dry process, there is less familiarity with its development history, the recent state of the research, or the mechanism of function of the PTFE binder. Furthermore, since the PTFE binder has generated great advancements in the SF technology, many researchers have become deeply interested in studying it, hoping to obtain details about the PTFE binder through more studies. Accordingly, this study provides an introduction into the current common SF processes; then explores the evolution of PTFE binders frequently used in the polymer fibrillization method. It also explores the characteristics of PTFE fibrillization and focuses on the review of the latest progress in several research directions regarding the PTFE-based SF process; finally, it summarizes the existing key problems concerning PTFE binders and predicts its future development trends in order to provide inspiration for subsequent researchers.

## 2. Developments in SF Processes and Binders

Over the past decade, SF processes have attracted great attention from both academia and industry. Yang Zhang et al. [26] summarized the number of papers published using the keywords “dry LIBs or solvent-free LIBs” (in the Web of Science) and the number of patent applications for LIBs using spray deposition and polymer fibrillation from 2006 to 2022, as shown in Figure 2a,b. Both the number of articles and patents show an overall increasing trend. In addition, Yongxing Li et al. [31] calculated the number of patents for polymer fibrillation and spray deposition methods in energy storage found in the global Derwent DWPI, CNABS, and Google Patents databases from 2022 up to 256, resulting in 140 and 116 patents, respectively (Figure 2c). These patents come from China, the United States, Japan, Germany, Korea, and other countries, as shown in Figure 2d. There were 59 polymer fibrillation cases out of 77 patents issued in the United States (Figure 2e), and 49 out of 57 patents issued in Japan involved the spray deposition method.

### 2.1. SF Processes

With the progressive realization of the significant advantages of the SF process, various SF electrode processes have evolved and have been employed in the fabrication of LIBs. As shown in Figure 3a–f, there are six typical SF electrode manufacturing processes: dry spraying deposition [11,27,33], vapor deposition [34,35], melting and extrusion [36], 3D printing [37,38], direct pressing [39,40], and polymer fibrillation [41,42,43]. The principles and applications involved in each process are presented below.

I.Dry spray deposition

As shown in Figure 3a, the method involves the dry mixing of the active material and the conductive agent first; then, the composite powder obtained is added to the spraying device. The composite powder (active material, conductive agent, binder) is sprayed onto the collector through the spraying device and then reinforced using the heat pressing method. For this method, the PVDF binder, or PVDF binder with a certain percentage of PTFE added, are commonly used [31]. Since most of the common active materials in LIBs are compatible with this method, it is currently more widely used in dry electrode manufacturing. However, the technology suffers from challenges regarding process upgradability and the inability to control the precise thickness. In addition, it is incompatible with the equipment currently used in LIBs production lines, and its efficiency is not comparable to that of the wet process. 

II.Vapor deposition

This method vaporizes and deposits materials onto a substrate, and the process includes atomic layer deposition, magnetron sputtering, and pulsed laser deposition [34,35]. This method can prepare electrodes with a high energy density and a long cycle life. However, the equipment required for this method is complex, and the operating environment requires the use of a vacuum. Therefore, this method is suitable for the manufacture of small-sized electrodes, and is not suitable for the large-scale manufacture of electric vehicle electrodes.

III.Melting and extrusion

This method is suitable for the preparation of a solid polymer electrolyte, which is widely used in the melt mixing of thermoplastics or ceramic slurry mixing. Although this method can prepare highly loaded electrodes, the process is sensitive to particle size and requires precise control of the extrusion temperature, shear force, and extrusion time. In addition, this method precludes its application in industrial manufacturing due to the cumbersome manufacturing process and the high de-binding and sintering temperatures.

IV.3D printing

This method fabricates electrodes using a fused deposition modeling (FDM) mechanism in which layers of molten polymer material are stacked horizontally using a specific nozzle tip. The freestanding electrodes produced by this method exhibit a specific morphology, and the thickness may be customized according to the specific application situations. Nevertheless, this technique cannot be applied to fabricate electrodes on a large scale; instead, it can only be used for specific scenarios, such as in the production of microelectronics and wearable devices.

V.Direct pressing

Direct pressing is also known as powder compacting, which is a direct method for pressing powder into shape. It is appropriate for preparing electrolytes and electrodes for all-solid-state batteries. As shown in Figure 3e, this method is simpler and more efficient compared to other methods. Nonetheless, this method exhibits the disadvantage of showing uneven stress and density distribution during unidirectional pressing. Since this method leads to a decrease in electrode density and an increase in porosity, it is not suitable for large-scale production.

VI.Polymer fibrillation

The process steps include four steps: (i) dry powder mixing, (ii) binder proto-fibrillation, (iii) self-supporting membrane molding, and (iv) self-supporting membrane composite formation, with a collector [31]. In this method, a polymer binder capable of forming a fibrous structure under the action of shear force is employed to achieve an effective connection between the electrode active particles and the conductive additives and other substances. The PTFE possesses significant fibrillar characteristics and highly stable thermo-mechanical properties. Therefore, most of the processes select PTFE as the binder in the polymer fibrillation method. In Figure 4, common proto-fibrillating equipment, which can a provide high speed of shear force are, i.e., an air flow mill, a screw extruder, a roller mill, etc. [31], are presented. Only after adequate proto-fibrillation could the binder form a self-supporting film with a more complete polymer network, reducing the electrochemical impedance. Maxwell showed that the resistance of fibrillated dry powders increases with a decrease in feed and injection pressures and also decreases with increasing grinding pressures [44].

In addition, two common electrode molding processes [45] include powder extrusion molding and powder roll molding, as shown in Figure 5a,b. Powder extrusion molding refers to the micro-fibrillated mixed powder created using a twin-screw extruder to increase high-speed shear to create the fibrillated binder and form it into a self-supporting film and a collector for the composite. Powder roll forming refers to the fibrillation of the mixed powder using a multi-stage roller press incorporating a differential shear roller pressing process so that the binder fibrillation and preparation of the self-supporting membrane can occur, followed by the production of the composite with the collector [45].

### 2.2. Binders

Actually, the SF process shows a tight relationship with the binder. The battery binder is a class of polymer compounds that adheres the active materials and conductive agents in the electrode sheet to the electrode collector, and it is one of the essential constituent materials of LIBs [13,46]. Although the amount of binder is small (∼5 wt%) [47], it serves to enhance the contact between the active material, the conductive agent, and the collector, as well as to stabilize the structure of the electrode sheet, which determines the change of the fabrication technology. Typical binders used in LIBs electrodes include PVDF [48,49,50,51,52,53], PTFE, styrene-butadiene rubber (SBR) [54,55,56], sodium carboxymethylcellulose (CMC) [57,58,59], poly (acrylic acid) (PAA) [60,61], poly (ethylene oxide) (PEO) [62,63,64], alginate [65,66,67], etc. The mechanical and thermal properties, etc., of the above binders are shown in Table 1.

In 2023, Tian Qin et al. [68] summarized the adhesion, tensile strength, elasticity, swelling, conductivity, thermal stability, and oxidation stability of seven types of binders. Among them, PVDF, as the major binder for commercial battery systems (cathode), shows the most satisfactory balance between the material and electrochemical properties, as shown in Figure 6. However, PVDF is not suitable for electrodes with high mass loading (>20 mg/cm^2^), high voltage, and large volume changes due to its low electron cloud density and relatively weak van der Waals force interactions. Currently, researchers are attempting to replace the traditionally used PVDF with binders such as CMC and alginate. The above categories of binders can allow for the replacement of NMP with water as a solvent for electrode preparation, and there are papers reporting that the performance of electrodes fabricated from water-based pastes is comparable to that of electrodes fabricated using NMP.

The mechanical properties of CMC, PAA, and alginate are inferior to those of PVDF; however, they are rich in polar groups such as carboxylic or hydroxyl groups. The free carboxylic acid groups are able to interact with the hydroxyl groups on the surface of materials such as silicon/carbon and aluminum foils, resulting in better bonding properties. Even though CMC is inexpensive and thermally stable, it exhibits high rigidity and brittleness. In order to resolve these issues, CMC is often used as an anode binder in conjunction with SBR, which has high elasticity. The silicon anode with the SBR-CMC composite binder showed a smaller Young’s modulus and stronger adhesion strength to the collector [69]. PTFE, SBR, CMC, PAA, etc., could be used with water as a solvent to reduce the toxicity associated with the use of organic solvents, whereas further optimization of the time-consuming and energy-consuming drying step is still necessary in order to decrease the cost of battery manufacturing.

Based on the above considerations, researchers have continued to explore new electrode fabrication methods that do not require the use of solvents. Spray deposition and polymer fibrillation are considered as the two most mainstream methods for the fabrication of SF electrode membranes [31,70]. PVDF is mainly used as binder in the spray deposition method [26], and the binders of polymer fibrillation include PTFE, ethylene-tetra-fluoro-ethylene (ETEF), and fluorinated ethylene propylene (FEP) [31]. The properties of these three materials in terms of melting point, dielectric constant, tensile strength, and static friction coefficient are shown in Table 2. A comparison of these properties reveals that PTFE exhibits better chemical stability, mechanical properties, and oxidative stability. The current best choice for use in the polymer fibrillation method is PTFE.

## 3. Binders of PTFE

Since the 1980s, binder materials have undergone several changes. Before the 1990s, “aqueous binders of PTFE “ were frequently utilized [73]. Nevertheless, this type of binder exhibited inadequate compatibility with the active substance. In 1991, Sony released an LIB using PVDF as the binder and NMP as the solvent, a combination which showed excellent electrochemical properties. Since then, battery companies like Daikin Industries (Japan) have focused on developing PVDF binders. Currently, most electrodes using PVDF as a binder use the SC process. Nevertheless, the rapid evaporation of the solvent NMP could consume more than 51% of the total energy required for battery manufacture [16]. Meeting the urgent requirements for the green production of batteries is compulsory to create a functional binder for triggering the emergence of a new battery fabrication process.

Tesla has explored an SF process using fluoropolymer PTFE as a functional binder. Polymer networks could be formed by PTFE fibrillation to prepare self-supporting electrode films. This process does not require a drying process or solvent recycling, which significantly reduces the cost of manufacturing electrodes and appears to be friendly to the environment. The PTFE binders once again in use in the battery field, thanks to Tesla’s research and development. PTFE have become the focus of both scientific study and business investigation. It could be classified as either SC or SF, depending on the processing procedure. The subsequent section provides a comprehensive overview of these procedures.

### 3.1. Aqueous Binders of PTFE

Currently, most cathode binders use the PVDF [46]. However, the NMP solvent is expensive and harmful to organisms and the environment [74]. Water-based binder performs equally as well as, or even better than, oil-based binder. Choosing the right amount of water-based binder could improve the behavior of the battery. Therefore, many companies are actively exploring the use of cheaper and more environmentally friendly water-based binders to replace PVDF. The PTFE binder shows extraordinary suitability for LIBs due to its excellent mechanical properties, electrochemical ability, and electrolyte compatibility [32]. In addition, PTFE could form an emulsion in aqueous solution containing stabilizers. In recent years, several studies have been conducted using aqueous bonding agents for PTFE. For instance, Gao et al. [75] applied a PTFE aqueous binder in the creation of C/LiFePO_4_ batteries. The SEM images of electrodes prepared using PTFE and PVDF are shown in Figure 7a,b. These batteries showed superior first-time Coulomb efficiency compared to that of PVDF. In 2018, S. Priyonol et al. [76] used PTFE to prepare wet electrodes of Li_4_Ti_5_O_12_. The battery heated to 40 °C before slurry coating on copper foil exhibited the highest specific capacity.

### 3.2. SF Binders of PTFE

Polymer fibrillation requires materials with unique characteristics. Currently, the most optimal selection for a binder is limited to PTFE due to its exceptional mechanical qualities, high crystallinity (achieving 97% or more post-sintering) [77,78,79], and its ability to produce fibers. PTFE exhibits flexibility and reduced resilience compared to the properties of other polymers, with moderate tensile strength and high elongation at the break [80,81]. The material undergoes deformation under specific pressure conditions, while retaining the correct dimensions. The above behavior perfectly matches the processing requirements for the SF method of electrode stretch molding. Thanks to these material properties, PTFE has become the most popular binder in the polymer fibrillization process for SF manufacturing applications. The properties of PTFE fibers are intrinsically linked to the inherent structure of PTFE. In the following section, we present a detailed analysis of the molecular structure and the principles of polymer fibrillation.

#### 3.2.1. Molecular Structure

PTFE possesses the highest chemical resistance, a high dielectric constant, and a wide range of operating temperatures [82]. The properties of PTFE result from its high crystallinity, high molecular weight, and unbranched structure. The radius of the F atom in PTFE is more significant than that of the H atom, and the C-F bonding energy is relatively vital (485 kJ mol^−1^) [80,81,82,83]. Therefore, the adjacent -CF_2_- units in the molecular chain structure cannot present a transverse cross-orientation conformation, as does polyethylene, and instead appear to be helically arranged in the overall conformation [84]. This spiral arrangement of the F atoms surrounds the carbon main chain. It covers the entire surface of the molecular chain, forming a protective layer of low surface energy around the C-C main chain. This non-polar and inert dense layer produces high intermolecular van der Waals repulsion. Macroscopically, PTFE exhibits excellent non-stick properties, low coefficient of friction, chemical and thermal stability, and many other properties [85].

#### 3.2.2. The Principle of Polymer Fibrillation

PTFE SF binder particles show an average particle size of 500 μm, with many oblate spheroidal particles. These particles are comprised of several folded lamellar crystals (0.54 μm long, 0.25 μm wide) [82,86]. When applying a shear load to PTFE, the oblate spheroidal particles exhibit flexibility, allowing them to elongate and form a fiber. This process is called “fibrillation” in Figure 8a,b [87]. 

Why can fibrillations occur using PTFE? There are two primary factors contributing to this phenomenon. One is the dislocation slip in the PTFE polymer crystals [28]. The polymer’s crystal structure determines the fibrillation behavior caused by dislocations. As shown in Figure 8c, the crystal structure of PTFE can be divided into four phases [29,88]: pseudo-hexagonal crystal (Phase I), trilobal crystal (Phase II), planar zig-zag crystal (Phase III), and hexagonal crystal (Phase IV). Taking the hexagonal crystals of Phase IV as an example, the pre-fibrillation operation in the range of 19–30 °C transforms the crystalline phase of PTFE from Phase II to Phase IV. The PTFE’s repeating distance along the molecular axis increases from 1.65 nm in the triple-diagonal crystals to 1.95 nm in the hexagonal crystals. Additionally, the repeating helical structure in the molecular chain expands -CF_2_- from 13 to 15, accompanied by a slight expansion in the structure of the helical repeating units [88]. The structure of the helical repeating unit is slightly expanded in Figure 8d [89]. In this phase of PTFE, the cohesion between the neighboring chains is weak and easily dislocated. In order to avoid polymer chain breakage, dislocations can generally only slide along planes parallel to the polymer chains, most commonly observed as chain slip and lateral slip. The sliding deformation of PTFE along the chain (c-axis of the hexagonal system) has been reported to be easier to deform into nanofiber structures under a high aspect ratio, as shown in Figure 8e [89].
Figure 8(**a**) PTFE spheres are stretched into banded fibers [87]. (**b**) SEM image (2 μm) of PTFE fibrillation [87]. (**c**) Phase diagram of PTFE [88]. (**d**) The individual PTFE polymer chain with helical structure and its simplified cylinder model [89]. (**e**) The PTFE crystals with slip dislocations occurring under shear [89].
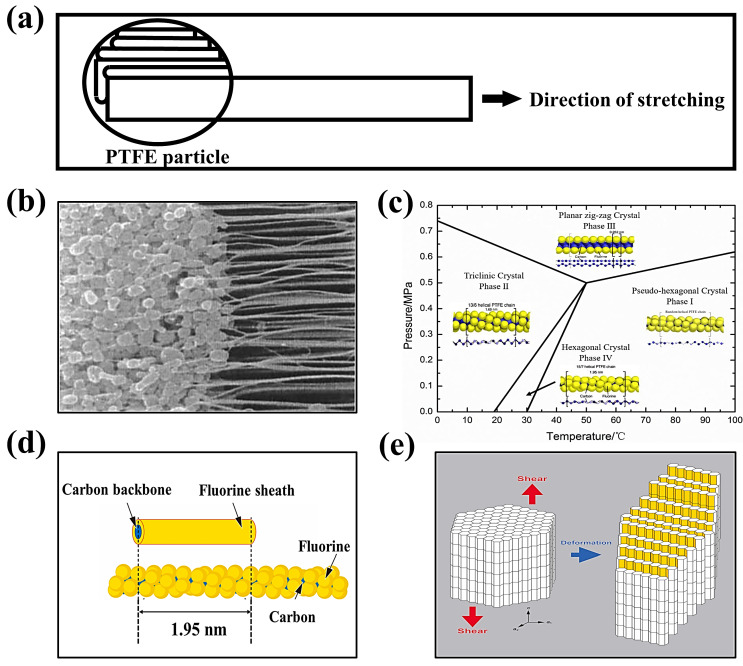



Secondly, the van der Waals repulsive force between molecules is strong in PTFE [89]. The significant electronegativity of the F atom allows for easy absorption of the electrons, reducing the intermolecular gravitational force. As a result, PTFE slips easily in the direction of external pressure. According to this feature, PTFE can be pre-fibrillated in the SF process up to a specific strain level, and the active materials and electrical agents will be evenly spread out. This process is macroscopically manifested as a fluffy mass, facilitating further efficient preparation of the electrode layer using shear roller pressing.

#### 3.2.3. Factors Affecting PTFE Fibrillation

Currently, there are fewer studies dedicated to discussing the influence of fibrillation properties. Thus, this paper also refers to the research on PTFE fibrillation applied to other fields. In the field of SF batteries research, Maxwell’s experimental data show that the impedance of the original fibrillated electrode film is related to the feed rate and the shear force [44]. Related reports in other fields also assist in investigating the mechanical behavior of PTFE. Aimin Zhang et al. [90] explored the fibrillation mechanism, crystallization behavior, and mechanical properties of in situ fiber PTFE-reinforced PP composites. The experimental results showed that the shear rate is the key parameter affecting the morphological evolution of PTFE, and the processing time also affects the morphology of PTFE, to a certain extent. Jennifer L et al. [91] chose the Zerillie Armstrong polymer model [92] to study the properties of the strain rate of DuPont Type 9B PTFE. In addition, the group tested the material using dynamic mechanical analysis and determined that time and temperature superposition affect the PTFE strain rate. Li et al. [93,94] found that the orientation structure of PTFE has a strong anisotropy effect on the tribological properties of tensile PTFE. Orientation is one of the important factors affecting the tribological properties of polymers, and the linear molecular chains of PTFE are easily oriented along the stretching direction. Through molecular dynamics simulations at the atomic level, Sinnott et al. [95,96] confirmed that the orientation of the molecular structure at the sliding interface of PTFE has a powerful effect on frictional wear. Furthermore, temperature also affects the deformation of PTFE. At atmospheric pressure and temperatures below 19 °C, shear forces cause PTFE particles to slide against each other. Above 19 °C, PTFE exhibits a first-order transition from a triclinic to a hexagonal crystal structure [97,98]. The cohesion between neighboring chains in the hexagonal crystal of PTFE is not strong enough [97,98]. As a result, the PTFE molecules stack more loosely, and shear leads to microcrystalline unfolding and the production of fibers. In particular, a high degree of fibrillation could be achieved at temperatures higher than 30 °C [99,100].

The influencing factor of PTFE deformation may consist of a single variable or a synergistic effect of multiple variables. As a binder in polymer fibrillation, PTFE should possess a smaller particle size and a higher molecular weight. Currently, the downstream application of PTFE is mainly concentrated in plastic products. However, research regarding PTFE applications for 5G communications and SF electrodes is still in the initial stages. Common manufacturers include DuPont, Daikin, etc. The properties of PTFE depend on the type of resin and processing method chosen. In other words, PTFE’s molecular weight, particle size, and crystallinity all influence the resulting physical and chemical properties. Researchers should follow up with a detailed investigation of battery-grade PTFE in terms of the mechanical properties (tensile strength, compression deformation, impact strength, and friction), the electrical properties (dielectric constant, electrical insulation strength, resistance value), and the thermal properties (coefficient of thermal expansion, thermal conductivity, thermal stability) [91].

## 4. SF Process with PTFE Binder

Over the past five years, there has been a significant increase in the use of SF processes in LIBs. For instance, in 2017, Maxwell Technologies reported on a highly loaded PTFE-based SF electrode technology named dry battery electrode (DBE) [44] that attracted the attention of researchers. In 2008, this company developed a fiber-able polymer to be used as a binder to prepare supercapacitors. Tesla acquired Maxwell in 2019 and released a new “4680” battery in 2020, with electrodes up to 250 um thick and a surface capacity of 6 mAh cm^−2^. Since Tesla obtained numerous patents for the “4680 cylindrical battery”, this company has occupied a leading position in fibrillation. Besides Tesla, many other universities and companies are also investigating SF methods. These include the Bosch Corporation (Farmington Hills, MI, USA), the China Hive Energy Company (Changzhou, China), the Technical University of Dresden (TU Dresden, Germany), the Toyota Corporation (Toyota, Japan), etc.

### 4.1. Positive Characteristics

Figure 9a shows a comparison of the wet and dry processes, and Figure 9b shows the SEM images of the electrodes for the wet and dry processes (polymer fibrillation). Polymer fibrillation stands out in the SF approach to batteries because of the following five advantages.

I.It is environmentally friendly and suitable for large-scale production.

NMP solvent is toxic, unfriendly to the environment, and needs to be recycled using the traditional wet process. The SF process does not require solvent in the electrode coating process to reduce baking and solvent recovery, the process is simpler, the equipment covers a smaller area, and the method is more suitable for the large-scale production of electrodes.

II.It exhibits a flatter electrode shape than that from the wet process.

Because the wet method requires solvent, after the solvent evaporation, the active substance and conductive agent will leave more spaces between the gaps, leading to the low compaction density of the material. The SF method does not exist in the drying process, so there is no solvent evaporation left after the gap, and the contact between the particles is closer.

III.It offers greater compaction density.

After compaction under dry conditions, there are fewer cracks, micropores, and other problems. The compacted density of lithium iron phosphate and SF battery energy density may be improved. According to Maxwell’s experimental data [44], the energy density of the SF electrode can be more than 300 Wh/kg, and has the possibility to realize 500 Wh/kg.

IV.It improves the performance of the battery

In the wet process, after the battery has gone through many cycles, the stresses within the active particles continue to accumulate, leading to cracks in the profile, which ultimately reduces the performance of the battery. In the SF process, the fiber network is wrapped around the surface of the active material, and the mesh structure remains intact after many cycles of charging and discharging. There are fewer cracks on the electrode surface, which enhances the stability of the battery.

V.It allows for the possibility of prepare solid-state batteries.

Empowered by SF technology, the manufacturing process for creating solid-state battery electrodes can be completely dried, eliminating the problem of solvent molecules remaining after drying in the wet process. In addition, the use of the original fibrillation manufacturing solid electrolyte film can reduce manufacturing costs so that solid-state batteries can also be more productive.

### 4.2. Development Status of SF Process with PTFE Binder

Due to the benefits from the SF process, PTFE binder has once again become a research hotspot in the energy field. Due to abovementioned advantages of PTFE, the primary studies regarding SF electrode technology are increasing. Among them, the efforts focusing on the PTFE SF procedure mainly center on the following aspects: PTFE, the other electrode components of non-PTFE, and novel SF technologies and systems. In this paper, we summarize reports regarding the PTFE solvent-free manufacturing process and then provide a detailed description of the recent development of the PTFE solvent-free procedure in the following sections.

#### 4.2.1. Effect of PTFE on SF Batteries

The properties of PTFE determine the performance of the solvent-free batteries and influence the SF process at the fundamental level. Relevant studies have been devoted to exploring whether changing the PTFE has an impact on the battery system, and the standard variables include the side reactions of positive and negative electrodes, ultra-low content (0.1–0.5 wt%), crystallinity, modified molecules of PTFE, substitutes for PTFE, and synergistic polymer binder.

I.Side reactions of PTFE binders

The reaction between PTFE and Li^+^ on the surface of the negative electrode will preferentially react to generate lithium fluoride, weakening the bonding effect and even destroying the PTFE electrode fiber network, leading to a rapid decline in electrode performance. This phenomenon was reported by Wu et al. in 2019 [102]. The chemical reaction between PTFE and Li^+^ can cause a low initial reversible capacity (<70%) of the anode of the LiNi_0.6_Mn_0.2_Co_0.2_O_2_ (NMC622)/graphite full battery. Chen’s group raised This problem as early as 1996 [103]. As shown in Figure 10a, this group found that PTFE would have a side reaction with Li+ on the surface of the carbon electrode at a low potential during the cyclic-voltammetry curve test of the PTFE/carbon fiber battery. An irreversible lithium depletion occurs during the first discharge. To further explain this problem, Salini et al. [104] discussed the HOMO and LUMO energy levels of standard polymer binders. As shown in Figure 10b, both energy levels of PTFE are relatively low. Its strong electron-gaining ability makes it more suitable for use as an anode binder. The above results demonstrate that PTFE possesses a superior ability to bind to Li+ compared to that of other materials at the level of theoretical calculations.

Does PTFE reveal a similar problem in the cathode? In 2023, Tao et al. [105] compared the changes in the batteries’ cathode electrolyte interface layer (CEI) using electrolytes containing LiPF_6_ or LiClO_4_. Using LiClO_4_ can eliminate other possible F sources, thereby probing the decomposition of PTFE. As shown in Figure 10c, the CEI layer using LiPF_6_ as the electrolyte salt is much thicker than that in the LiClO_4_-based SF electrode. The above results confirm that PTFE also undergoes reactions in the cathode. This group investigated, for the first time, the distribution of the CEI layer in PTFE solvent-free batteries and also explored the decomposition process of the PTFE binder. However, the effect of this process on the positive electrode still requires further study.

How could these side reactions be resolved? In 2019, Hippauf et al. [106] attempted to reduce the PTFE content when preparing SF electrodes. With the reduction of the PTFE dosage, the occurrence of lithium depletion in the PTFE/carbon nanofiber (CNF) anode during the first discharge gradually diminished. In addition, in 2022, Zhang et al. [107] prepared a graphite anode containing these two binders by utilizing the synergistic effect of PTFE and PVDF. PTFE was used as a processing aid for preparing self-supporting electrode film, while PVDF was employed as a functional binder. PTFE belongs to the class of insulating polymers. When the amount of PTFE is excessive, it will affect the conductivity of the electrode, specifically increasing the impedance of the battery. Reducing the content of PTFE, while using a synergistic binder, is a promising method for solving the above problems. Alternatively, researchers can develop a novel fluorinated viscosity modifier with more desirable HOMO and LUMO energy levels. For example, a higher HOMO energy level than that of PTFE makes its reduction under negative electrode applications difficult and prevents side reactions from occurring [103,104].

II.Crystallinity

In addition to the PTFE content, the crystallinity also affects the SF procedure. To investigate the effect of PTFE crystallinity on all-solid-state batteries (ASSBs), a class of sulfide-based Li_6_PS_5_Cl -ASSBs was constructed by Dongsoo Lee [86] in 2023. The PTFE formulations used in this experiment were non-crystalline (18.8%), semi-crystalline (41.3%), and highly crystalline (88.1%), respectively. The high crystallinity PTFE exhibited more substantial mechanical properties and closer contact with the active substance particles. Because it promotes uniform charge transfer in the battery, this PTFE can significantly improve the performance of ASSBs.

III.Modified materials

Battery-grade PTFE exhibits the limitation of being difficult to store. Previous studies have attempted to modify or replace PTFE. In 2022, Hong et al. [108] developed a modified PFTE material: poly (tetrafluoroethylene-co-perfluoro (3-oxo-4-pentanesulfonic acid)) lithium. The cross-sectional SEM images of the cathode without adhesive, containing PTFE and the ionic polymer, are shown in Figure 11. The ionic polymer exhibits good adhesion properties and high ionic conductivity. Although this material cannot fibrillate, its viscoelasticity allows for the uniformly dispersed ionomer to provide a more continuous pathway for Li+ conduction. In the same year, Schmidt et al. [109] tried to locate an alternative binder to PTFE. Eventually, the group discovered a fluorine-free material that could also fibrillate—silk glue. Comparative testing of batteries, using both as binders, revealed similar properties. The experiment proved that silk glue represents a class of materials with the potential to replace PTFE.

#### 4.2.2. Influence of Components Other Than Binders

At present, SF process not only research PTFE as a variable, but also explore the performance of batteries with other components. These variables include conductive additives, electrode materials, electrolytes, collectors, and so on.

I.Conductive additives

For the variable of conductive additives, in 2023, Yang et al. [110] prepared a full battery, with different carbon active materials (graphite, stiff carbon, and soft carbon) as the negative electrode, and LiNi_0.5_Co_0.2_Mn_0.3_O_2_ as the positive electrode. As shown in Figure 12a, the hard and soft carbon negative electrodes exhibited better cycling stability than that of graphite due to their small volume expansion during charge storage, and this work successfully expanded the application scope of the PTFE-based SF process. Their group also investigated similar work, as shown in Figure 12b [111]. They prepared SF Li^+^ phosphate (SF-LFP) batteries using PTFE as the binder (5 wt%) and carbon nanotubes as the conductive additives. The SF batteries exhibited superior cycling stability because the carbon nanotubes could be used as a matrix to accommodate the LFP particles. In the same year, Gyori Park et al. [112] used different conductive carbons (CNT and CB) to explore the corresponding battery performance, and they prepared NCM811, CNT, and CB with PTFE-based SF batteries. The final results showed that the carbon nanotube electrodes exhibited the best performance in terms of the cycle test, capacity retention, and mechanical properties.

II.Electrode materials

The electrode material will not only have an effect on the SF electrode method alone, but it will also affect the overall capacity of the SF batteries. For SF electrode technology, measures such as developing electrode materials with higher specific capacity, increasing the proportion of active substances, and selecting active substances in the appropriate voltage range could enhance the overall performance of the SF batteries. Based on the method using the PTFE solvent-free electrode to adjust for the variation of other components, it will be expected to obtain higher performance batteries. This means that there is an excellent opportunity to prepare high-capacity batteries from the perspective of the PTFE solvent-free process.

III.Collectors

Although there are previous works reporting on collector-free LIBs, most of them focus on the design of an integrated battery structure concept. For SF batteries, current research shows that collectors are still necessary. Currently, the SF batteries usually obtained using the polymer fibrillation method use temperatures of 180 °C and above [] to hot press and hold electrode self-supporting film for a period of time, while employing collectors such as aluminum foil, copper foil, carbon coated aluminum foil, etc. In addition, it is observed that the SF batteries can employ mesh collectors such as titanium mesh, nickel mesh, etc., and using this method, the self-supporting film was pressed into the inside of the mesh holes. In 2017, Yan Jing et al. [113] used PBDTD(S)@CNT as the positive electrode. A mixture containing 60 wt% composite, 30 wt% super phosphorus carbon, and 10 wt% PTFE binder was hand ground using an agate mortar. The resulting membrane was pressed into a stainless mesh with a screen size of 400 × 400 and a wire diameter of 0.001. The batteries exhibited a capacity in excess of 200 mA h g^−1^, with excellent stability even after 250 cycles. In addition, the use of conductive adhesive to adhere the fluid collector to the self-supporting membrane may be also a suitable choice.

#### 4.2.3. Innovative Technology and System

Current research on PTFE-based SF process batteries is no longer limited to the material itself. The exploration of battery processes and systems is also essential. Nevertheless, several efforts have also focused on novel SF battery systems, such as those involving high-voltage, solid-state, and high-load battery systems.

I.High-speed airflow technology

As shown in Figure 13a, in 2020, Zhou et al. [114] used a high-speed airflow impact to defibrillate PTFE, along with the use of conductive additives and the inclusion of active materials. Subsequently, they successfully prepared SF-LFP electrodes by combining hot rolling and hot covering processes.

II.Lithium-sulfur (Li-S) batteries

In 2023, Magdalena et al. [115] achieved the preparation of SF all-solid-state lithium-sulfur batteries with high sulfur utilization (ASSB-LiS) by using a high-energy ball milling method. The group pioneered the monitoring of the thickness of ASSB-LiS electrode sheets prepared using the SF process. This approach enabled a deeper understanding of the charging and discharging behavior of ASSB-LiS.

III.High-voltage batteries

High-voltage batteries are crucial for the development of the sustainable LIBs market. Moreover, high-voltage cobalt-free batteries with high energy density and low cost-effectiveness offer new possibilities for the battery industry. In 2023, Yao et al. [101] developed a 5V-grade cobalt-free battery based on a PTFE-based SF process, which enabled the successful preparation of a highly loaded spinel-type oxide LiNi_0.5_Mn_1.5_O_4_ (LNMO) electrode. The wet electrode surface loading performance starts to decrease at 4.0 mAh cm^−2^. However, the battery electrochemical performance continues to remain stable up to 9.5 mAh cm^−2^ (240 μm thick) of the SF electrode. A two-dimensional model constructed using plasma-focused ion beam scanning electron microscopy (PFIB-SEM), as shown in Figure 13b, also demonstrates that the electron-permeable network in the SF electrode is more effective in supporting homogenized charge conduction.

IV.High-load batteries

In 2023, Tao et al. [116] constructed a whole battery consisting of a highly loaded graphite (6.6 mAh cm^−2^) as the negative electrode and LiNi_0.6_Mn_0.2_Co_0.2_O_2_ (6.0 mAh cm^−2^) as the positive electrode by using PTFE as the binder using the SF process in order to compare it with the wet approach. The SF procedure exhibits significant advantages over the SC process in regards to multiplication performance and capacity retention for the entire battery.
Figure 13(**a**) Procedure diagram of pre-fibrillated PTFE using the high-speed airflow impingement method [114]. (**b**) A 2D model showing current density in SF and SC processes constructed by PFIB-SEM [101]. (**c**) Preparation of a PTFE-based solid-state electrolyte and solid-state electrodes [117].
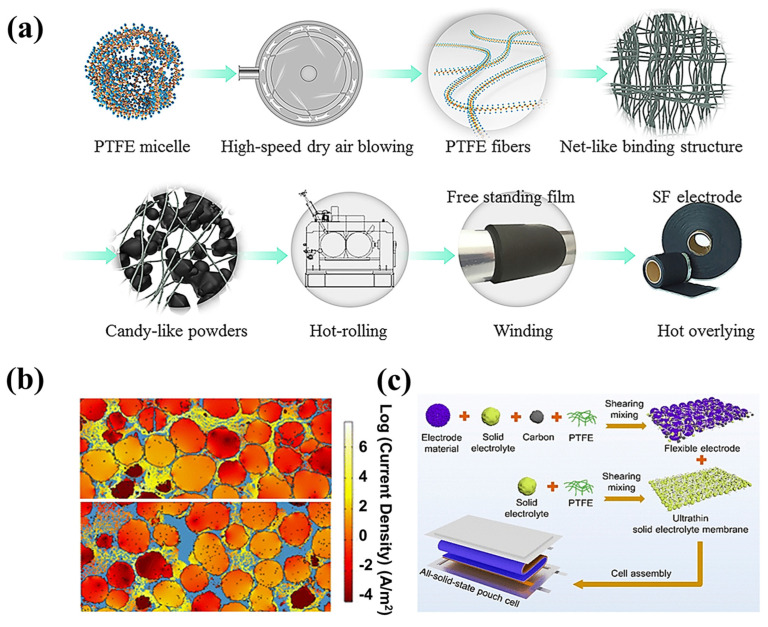


V.Solid-state batteries

The development of the SF-solid-state battery system should not be underestimated. Dong et al. [30] investigated the factors affecting the performance of SF solid-state batteries by exploring the content of the cathode active material and the solid electrolytes. The data from the batteries show that the most critical design criterion for SF solid-state batteries is the need for a reasonable and balanced conduction path in the SF composite electrode. Wang et al. [117] used the process shown in Figure 13c, using a blend of a 0.5 wt% PTFE and a 99.5 wt% solid-state battery. The ultrathin inorganic solid electrolyte films of Li_6_PS_5_Cl, Li_3_InCl_6_, and Li_6.5_La_3_Zr_1.5_Ta_0.5_O_12_ (LLZTO), with thicknesses of 15–20 μm, as shown in Figure 14, were prepared by roller pressing using 99.5 wt% solid electrolyte powder. Their room temperature ionic conductivity reached more than 1 mS cm^−1^. The energy density of the solid-state soft-packed battery assembled from LLZTO, NCM811, and Si-C-450 (8.02 mg cm^−2^) could reach 280 Wh kg^−1^, the highest level reported during that period. In addition, the capability of this soft-packed battery to supply energy to LEDs, even after being cut, also indicates that the SF-LLZTO solid-state electrolyte is a promising alternative to conventional electrolytes.

## 5. Challenges and Recommendations

The utilization of PTFE binder has addressed numerous problems in the battery industry. Nevertheless, using PTFE binder and the solvent-free technology of PTFE still presents considerable difficulties. In Section 3, we summarized some difficulties that need to be addressed, based on the information in the current literature.

These dilemmas present great challenges; therefore, we analyzed each one and offered recommendations, point by point, as follows: (1) We recommend the use of higher bonding requirements. Due to the high expansion coefficient caused by ionic de-embedding in the LIBs, they tend to be easily de-powdered after several cycles. Perhaps the bonding effect can be enhanced if the PTFE bonding agent is modified, or if PTEE is mixed with a non-primary fibrillated bonding agent, and if other bonding agents with smaller particle sizes are used. (2) We observe that PTFE reacts easily with negative electrode; thus, the LUMO orbitals of PTFE have lower energy and are prone to accept electrons, leading to a decrease in battery capacity. Coating a layer of conductive carbon on the surface of PTFE for passivation modification can weaken the reaction between PTFE and the negative electrode. (3) We recommend higher requirements for cathode rolling pressure. The electrochemical activity of the anode active material is relatively high. Therefore, chemical changes occur more easily in the process of roll pressing. Moreover, the self-supporting film of the cathode material after rolling is likely to shed powder. Perhaps the solution for the abovementioned problems may include increasing the additives, improving the electrode preparation, or changing the roll pressing equipment for better precision. (4) We suggest that the adhesive distribution should be uniform. Once the size or content of the PTFE adhesive fiber becomes relatively large, the adhesion effect will be weakened. The PTFE content can be reduced, or the equipment should be replaced with a device exhibiting a higher-pressure range. In addition, the rollers should be revolved several times during the rolling process. (5) We note a large internal impedance. The internal resistance of the batteries using the SF electrodes is higher, related to the following: a higher impedance at the interface between the aluminum foil and the SF active material; a possible polarization problems at high currents; and a higher solid-solid interface impedance in solid-state SF batteries. Measures such as using carbon coated collectors, replacing positive and negative electrode materials with those of a high specific capacity, reducing the thickness of the electrodes, and increasing the roll pressure may be considered.

## 6. Conclusions

In this paper, we introduced the development of PTFE-based SF and SC binders, analyzed the PTFE molecular conformation and fibrillation principle, evaluated the latest progress and challenges of the PTFE-based SF process, and provided a direction for the commercial production of SF electrodes. Optimized electrode binders for LIBs should exhibit excellent bonding capacity, perfect electrolyte compatibility, superior electrochemical stability, a wide redox window, excellent mechanical properties, competitive cost, etc. At present, PTFE remains irreplaceable as an SF binder. In the future, the crucial goal will be the development of new materials and devices, the creation of new methods of handling the pre-fibrillation, and the ability to accurately control SF self-supporting film production by employing laboratory research and large-scale production in this enterprise.

## Figures and Tables

**Figure 1 materials-16-07232-f001:**
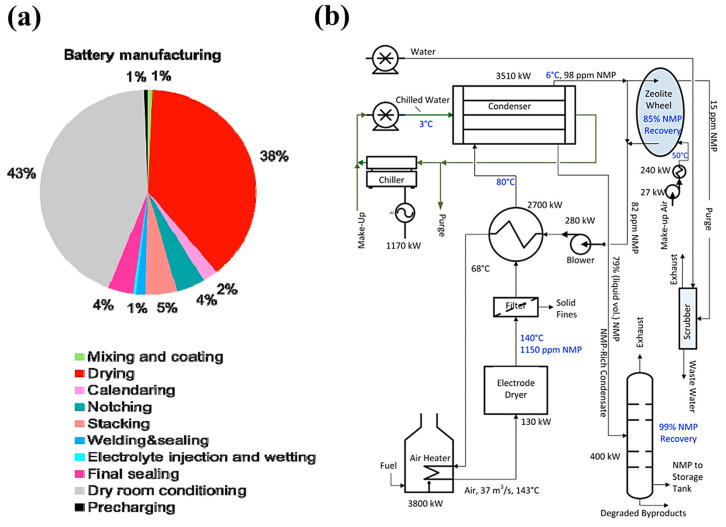
(**a**) Energy consumption of the LMO-graphite battery pack [16]. (**b**) Process schematic for the drying and recovery of the cathode solvent NMP [17].

**Figure 2 materials-16-07232-f002:**
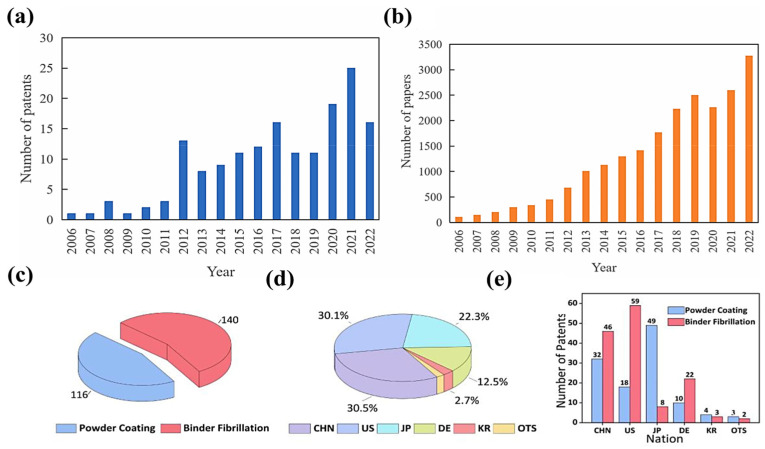
(**a**) The number of papers published with the keywords “dry LIBs or solvent-free LIBs” in the Web of Science from 2006 to 2022. (**b**) The number of patent applications for LIBs using spray deposition and polymer fibrillation from 2006 to 2022 [26]. (**c**) Patent distribution regarding binder fibrillation and powder coating. (**d**) Distribution of the number of patents filed from China, the United States, Japan, Germany, Korea, and other countries. (**e**) Number of patents filed per country for the two SF electrode manufacturing methods [31].

**Figure 3 materials-16-07232-f003:**
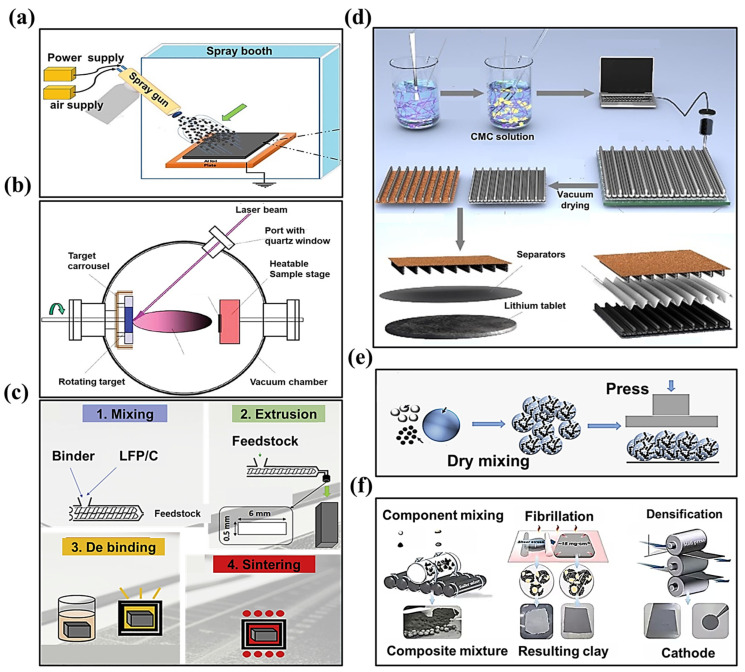
SF electrode manufacturing processes of (**a**) dry spraying deposition [33], (**b**) vapor deposition [34,35], (**c**) melting and extrusion [36], (**d**) 3D printing [37,38], (**e**) direct pressing [39,40], and (**f**) polymer fibrillation [41,42,43].

**Figure 4 materials-16-07232-f004:**
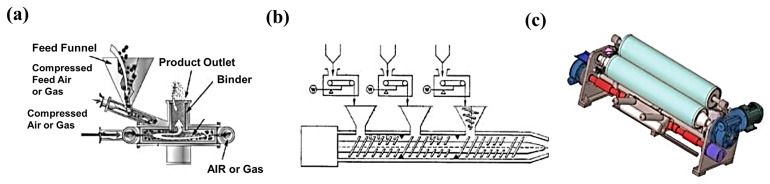
Proto-fibrillating equipment of (**a**) an air flow mill, (**b**) a screw extruder, and (**c**) a roller mill.

**Figure 5 materials-16-07232-f005:**
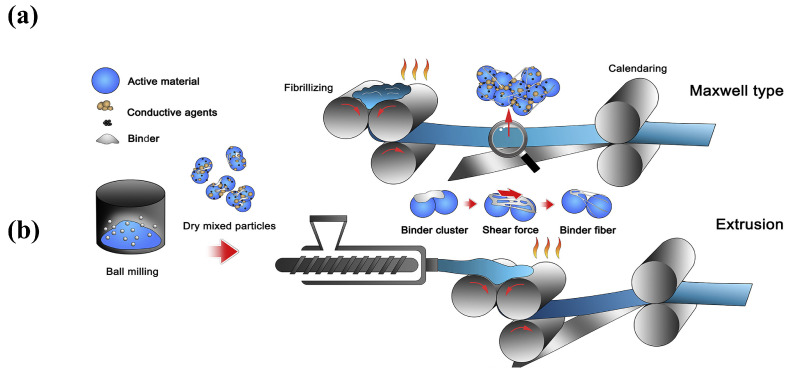
Processes of (**a**) powder extrusion molding, and (**b**) powder roll molding [45].

**Figure 6 materials-16-07232-f006:**
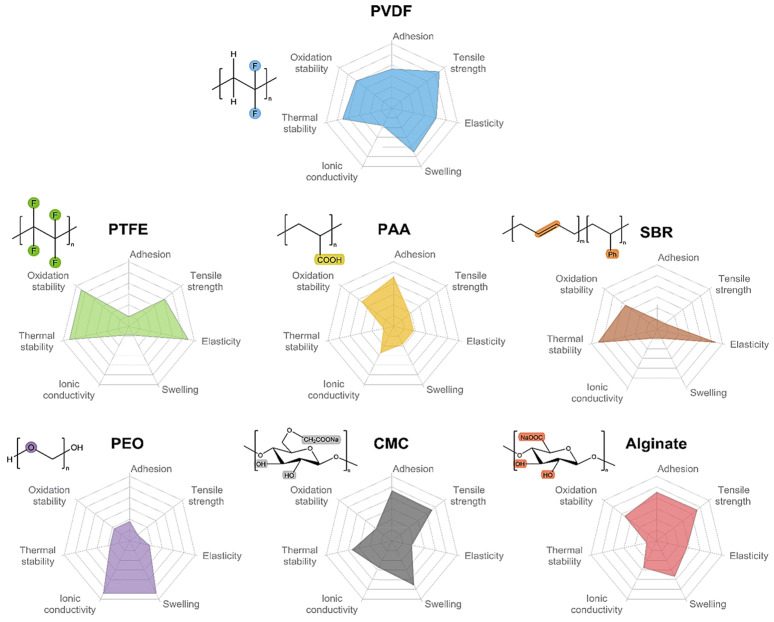
Comparison of the basic properties of common binders.

**Figure 7 materials-16-07232-f007:**
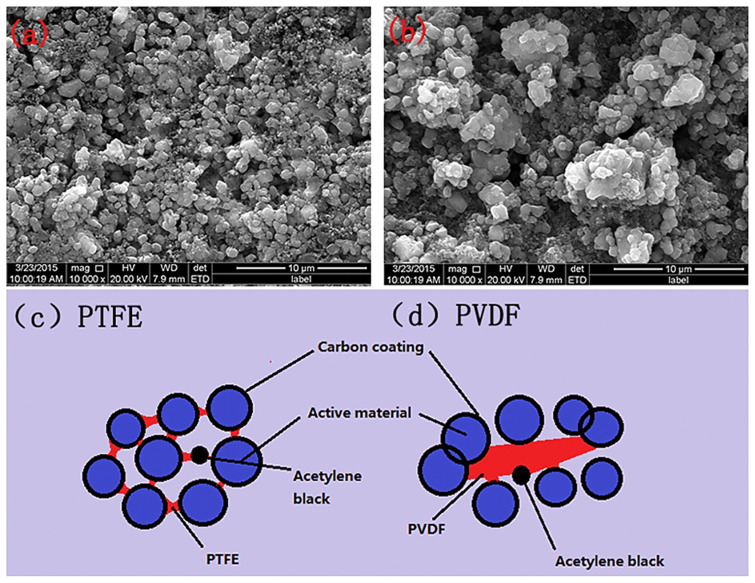
SEM image of LiFePO_4_/C electrodes prepared using (**a**) aqueous binders of PTFE and (**b**) PVDF, as well as the electrodes combined with (**c**) aqueous binders of PTFE and (**d**) PVDF [75].

**Figure 9 materials-16-07232-f009:**
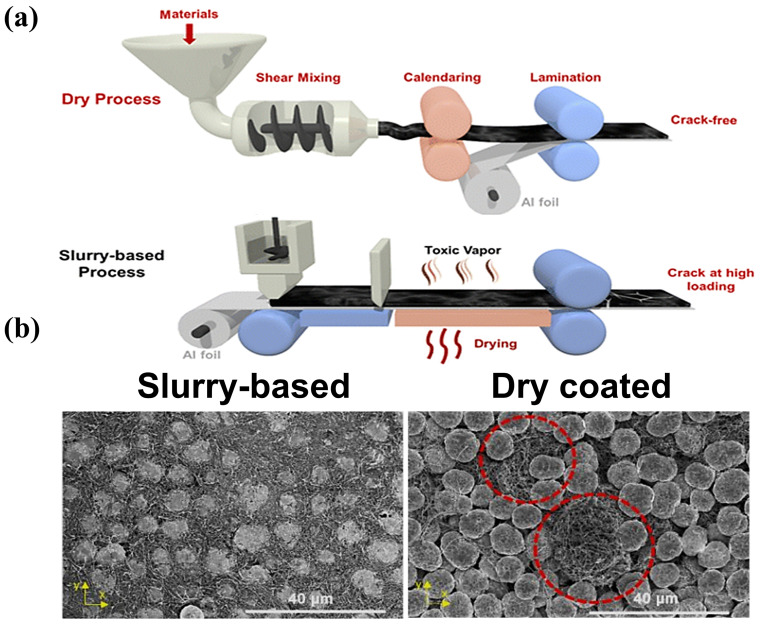
(**a**) Scheme of the PTFE-based SF and SC processes [101]; (**b**) the top surface of slurry-based LNMO electrodes and SF coated eletrodes [101].

**Figure 10 materials-16-07232-f010:**
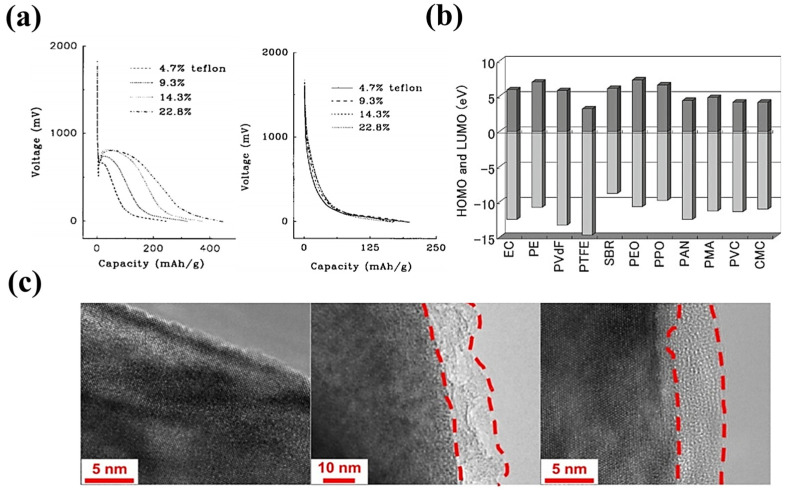
(**a**) The discharge curves of the first (left) and the second (right) of four different cells (Li/CF047, Li/CF093, Li/CF143 and Li/CF228). [103]. (**b**) HOMO and LUMO energy levels of various binders [104]. (**c**) Original and post-cycling TEM images of LiPF_6_ and LiClO_4_ electrodes [105].

**Figure 11 materials-16-07232-f011:**
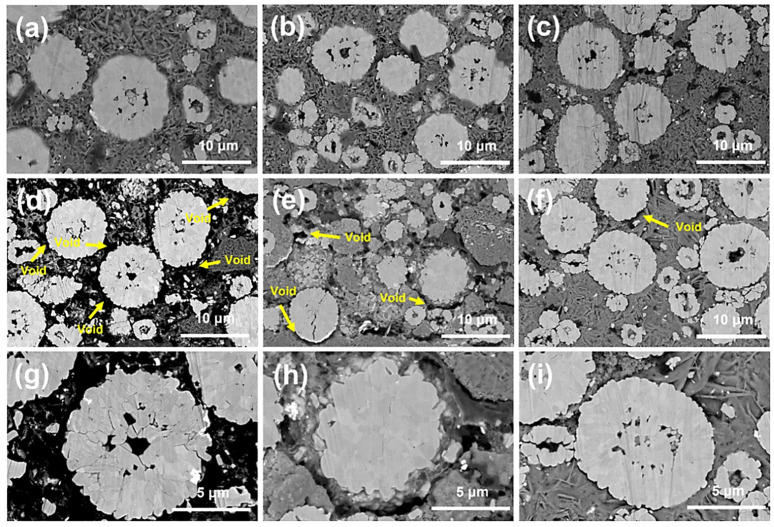
Cross-sectional SEM images of cathodes prepared (**a**) without binder, (**b**) with PTFE, and (**c**) with the ionic polymer. Cross-sectional SEM images of cycled (300 cycles) cathodes prepared (**d**,**g**) without binder, (**e**,**h**) with PTFE, and (**f**,**i**) with the ionic polymer [108].

**Figure 12 materials-16-07232-f012:**
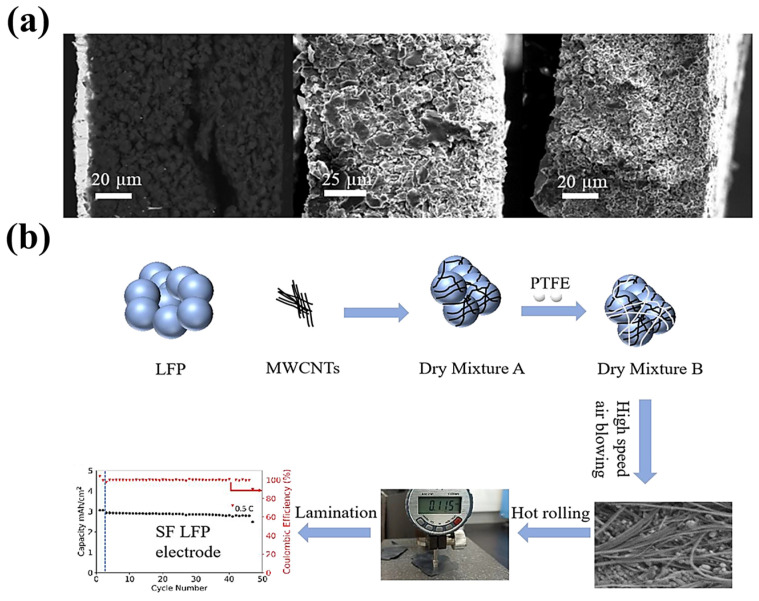
(**a**) Cross-sectional SEM images of a graphite, hard carbon, and soft carbon SF anode [108]. (**b**) SF-LFP electrode procedure [111].

**Figure 14 materials-16-07232-f014:**
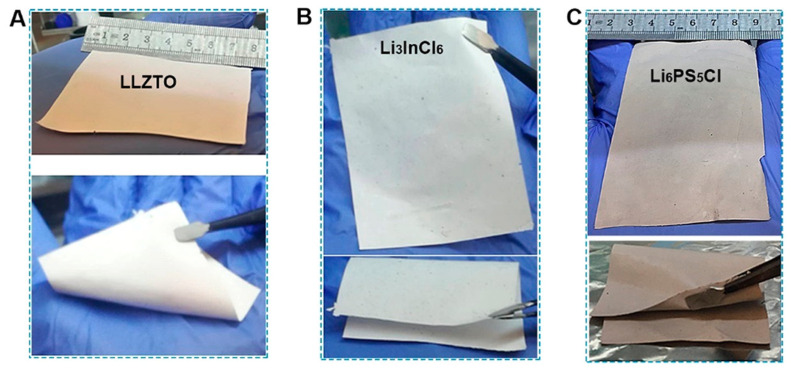
Characterizations of solid electrolyte membranes. (**a**) LLZTO. (**b**) Li_3_InCl_6_. (**c**) Li_6_PS_5_Cl [117].

**Table 1 materials-16-07232-t001:** Comparison of properties, i.e., thermal and mechanical behavior, of the binder.

Binder	Characteristic Functional Groups	Elastic Modulus (Mpa)	Break Elongation (%)	Ultimate Strength (MPa)	Adhesion Force (N cm^−1^)	Tg andd (°C)	References
PVDF	Fluorine	1400	30	40	1	−38,400	[48,49,50,51,52,53]
PTFE	Fluorine	400–1800	50–650	10–43	–	−103,400	[32]
SBR	Alkene, phenyl	1.31	385	3.33	0.1–0.5	−60,400	[54,55,56]
CMC	Carboxyl ion	1400	<10	40	1.1–1.7	55,300	[57,58,59]
PAA	Carboxyl	450	<5	<10	1.5	115,150	[60,61]
PEO	Ether	700	<10	15	<0.5	−50,350	[62,63,64]
Alginate	Carboxyl ion	1400	15	30	2	119, 200–500	[65,66,67]

**Table 2 materials-16-07232-t002:** Performance of binders for the fibrillation procedure.

Binders for Fibrillation	Melting Point (°C)	Dielectric Constant (°C)	Tensile Strength (Mpa)	Static Friction Coefficient	References
PTFE	327	2.1	25–40	0.02	[31,32]
ETEF	260–270	2.6	40–50	0.06	[71]
FEP	275	2.1	20–25	0.05	[72]

## Data Availability

Data are available in the source publications listed in the bibliography.

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
