# Peer review of "A Polytetrafluoroethylene-Based Solvent-Free Procedure for the Manufacturing of Lithium-Ion Batteries"

_materials, 2023, doi:10.3390/ma16227232_

Round 1

Reviewer 1 Report

Comments and Suggestions for Authors

Solvent free casting method is considered as a sustainable and next-generation battery fabrication technique which reduces the negative environmental impact and cost of production. Dry electrode fabrication using fibrillation polymers is a popular technique. Here in this review, the authors present the development of the Polytetrafluoroethylene (PTFE) based fibrillation polymer for solvent-free battery fabrication. Although the topic is highly relevant, the manuscript preparation is too vague. The organisation of the manuscript need improvement and the critical analysis is absent in the review. Many observations are repeated throughout the manuscript. Additionally, the perspective part of the review needs improvement. In conclusion, the manuscript is too early to considered to be for publication. The authors are requested to revisit the manuscript; there is much room for improvement. The editors are requested to give more than two weeks for the major revision of the manuscript.

1.      In the introduction, the author should discuss solvent recovery cost while using NMP, the total energy required for battery manufacture, etc. Also, the author should add how the solvent-free casting method can improve sustainable manufacturing.

2.      The author should revisit the Figures 1g and 1h. Most of the papers with the keyword “polytetrafluoroethylene binder” are not related to batteries. Also, the author claims that there is 1000+ papers related to “dry lithium-ion batteries” which are not true.

3.      What are other fibrillation polymers? What could be the advantage of PTFE over the other fibrillation polymers?

4.      Tabulate the properties of PTFE, PVDF and other binders like CMC-SBR like dielectric constant, voltage window, operational temperature, Cost, the Energy gap between HOMO and LUMO, etc in the introduction.

5.      In Table 1, the PTFE content,  crystallinity of PTFE, thickness of the electrode, the conducting agent, etc. should also be included. What rationale did the author take for choosing only seven previous reports in Table 1?

6.      What could be the critical shear load to be applied for fibrillation? The author should also add a section about the factors affecting fibrillation. The author could consider adding the non-battery papers which focus on the fibrillation of PTFE.

7.      The fibrillation of the PTFE is not clearly visible in Fig. 4b. The authors are requested to check other reports for better images.

8.      Section 3.2.2.1 The influence of components except binders (Electrode materials) needs further discussion on how the influent of different electrode materials are influencing. 3.2.2 section should also discuss the other components like electrolyte effect, current collector effects etc. Some of the previous reports show the current collector free some are with current collector. The author should add a rational discussion on this.

9.      The authors are recommended to avoid the repetition of the Tesla’s history. The authors must declare if there are conflict of interest.

10.   The authors can consider one more section on the application of the dry electrode fabrication progression in lithium sulfur batteries and solid state batteries which will make the review more impressive.

Comments on the Quality of English Language

The authors must improve the language for readability of the manuscript.

Author Response

Responds to the reviewer’s comments:

Reviewer 1:

Comments: Solvent free casting method is considered as a sustainable and next-generation battery fabrication technique which reduces the negative environmental impact and cost of production. Dry electrode fabrication using fibrillation polymers is a popular technique. Here in this review, the authors present the development of the Polytetrafluoroethylene (PTFE) based fibrillation polymer for solvent-free battery fabrication. Although the topic is highly relevant, the manuscript preparation is too vague. The organization of the manuscript needs improvement and the critical analysis is absent in the review. Many observations are repeated throughout the manuscript. Additionally, the perspective part of the review needs improvement. In conclusion, the manuscript is too early to considered to be for publication. The authors are requested to revisit the manuscript; there is much room for improvement. The editors are requested to give more than two weeks for the major revision of the manuscript.

Answer: Thank you for your comments.

Comments (1): In the introduction, the author should discuss solvent recovery cost while using NMP, the total energy required for battery manufacture, etc. Also, the author should add how the solvent-free casting method can improve sustainable manufacturing.

Answer: Thank you very much for your suggestion. We have reworked the introduction section with the content on the energy required for NMP recycling and battery manufacturing depletion in lines 40-57. In addition, we have re-discussed the SF process to improve sustainable production in lines 62-80 and section 4.1.

Comments (2): The author should revisit the Figures 1g and 1h. Most of the papers with the keyword “polytetrafluoroethylene binder” are not related to batteries. Also, the author claims that there is 1000+ papers related to “dry lithium-ion batteries” which are not true.

Answer: Thank you for your positive comments. Indeed, as you suggest, Figures 1g and 1h have no rational sense. We have re-quoted and rewritten this section. The rewritten portion can be found in lines 94 to 115 in the manuscript.

Comments (3): What are other fibrillation polymers? What could be the advantage of PTFE over the other fibrillation polymers?

Answer: Thank you very much for your question. We revisited previous work and found several classes of materials that also have fibrillating capabilities. Furthermore, we compared these materials in terms of their mechanical and thermodynamic properties, which are discussed in detail in lines 236 to 250 of the draft.

Comments (4): Tabulate the properties of PTFE, PVDF and other binders like CMC-SBR like dielectric constant, voltage window, operational temperature, Cost, the Energy gap between HOMO and LUMO, etc in the introduction.

Answer: Thank you for your positive comments. We do ignore the development of PTFE with other binders and the comparative performance between these materials. We believe that this section has a greater impact on highlighting the advantages of PTFE in the SF process. Therefore, we have analyzed in detail the seven most commonly used types of binders. The details can be found in lines 197 to 250 in the manuscript.

Comments (5): In Table 1, the PTFE content, crystallinity of PTFE, thickness of the electrode, the conducting agent, etc. should also be included. What rationale did the author take for choosing only seven previous reports in Table 1?

Answer: Thank you very much for your suggestion. Table 1 summarizes the data on PTFE dry batteries. However, we recognize that this table does not convey our intent to highlight the importance of the PTFE binder. Therefore, we have re-summarized Tables 1 and 2 to specifically address binder properties.

Comments (6): What could be the critical shear load to be applied for fibrillation? The author should also add a section about the factors affecting fibrillation. The author could consider adding the non-battery papers which focus on the fibrillation of PTFE.

Answer: Thank you very much for your question. We also recognize that our work is missing a section on the effects of PTFE fibrillation. To address this issue, we refer not only to articles in the battery field, but also to articles on PTFE fibrillation in other fields. The specific additions can be found in lines 353-391 in the manuscript. We are aware of a lot of work, but are still unable to give specific influences or critical values on this issue. These are also issues that the battery field and the polymer field need to continue to work on continuously to solve. We will continue to work on this issue in detail in the future as well.

Comments (7): The fibrillation of the PTFE is not clearly visible in Fig. 4b. The authors are requested to check other reports for better images.

Answer: Thank you for your positive comments. We have replaced the picture with a clearer image of fibrillated PTFE. The photo has now changed to the 9th diagram, which can be found on line 439 in the manuscript.

Comments (8): Section 3.2.2.1 The influence of components except binders (Electrode materials) needs further discussion on how the influent of different electrode materials are influencing. 3.2.2 section should also discuss the other components like electrolyte effect, current collector effects etc. Some of the previous reports show the current collector free some are with current collector. The author should add a rational discussion on this.

Answer: Thank you very much for your suggestion. We have added discussion of electrode materials and collectors to sections 3.2.2.1 and 3.2.2. For electrolytes, we have merged it with the chapter on solid-state electrolytes.

Comments (9): The authors are recommended to avoid the repetition of the Tesla’s history. The authors must declare if there are conflict of interest.

Answer: Thank you very much for your question. We have removed duplicate content. We have also taken conflict of interest seriously once again. In the revised manuscript, we made every effort to avoid this problem.

Comments (10): The authors can consider one more section on the application of the dry electrode fabrication progression in lithium sulfur batteries and solid-state batteries which will make the review more impressive.

 Answer: Thank you for your positive comments. We have added corresponding content in section 4.2.3. about lithium-sulfur batteries, solid-state batteries and so on.

Reviewer 2 Report

Comments and Suggestions for Authors

This is a review paper for a solvent-free procedure for lithium-ion batteries (LIBs). The article discusses the role of binders, mainly focusing on Polytetrafluoroethylene (PTFE), and highlights the transition from traditional slurry-casting (SC) procedures to a more cost-effective and solvent-free (SF) manufacturing method. The overarching goal is to inspire future research aimed at enhancing the quality of and advancing the evolution of the SF manufacturing process for LIBs. Below are some improvements to consider.

- Introduction. Please add a short discussion of why the current practices of manufacturing LIBs are not ecologically friendly. Add some statistics if possible.

- Also, provide the breakdown of the review paper. Line 76-79. In a new paragraph, please expand on how the review paper followed the PRISMA guidelines. What databases were used, the keywords, and how many papers were reviewed? Refer to Systematic Review: https://www.mdpi.com/about/article_types

- Lack of Methodology discussion: The review paper does not provide general details about the methodology used, the differences in the methodology process, or specific results obtained between polymer fibrillation and vapor deposition. Readers may be interested in understanding how the dry binder was developed and its performance characteristics. Suggest adding another section providing a brief overview of the methodology and key to give readers a better understanding of the paper's contributions.

- Section 3.2.2. Line 317. Please briefly describe before the bullet points to match with other sections. Similarly, Section 4. Line 394.

- Section 4.1 can be expanded rather than having seven bullet points. For example, why is the performance of semi-crystalline or non-crystalline solvent-free is undesirable (Point #3). Why preparing and storing high-molecular weight is complex? How complex? (Point #4). Why is high-quality, efficient quantitative production of electrodes tricky? What makes them tricky? (point #6)

- Section 4.2 suggests linking the recommendation to the challenges #. Section 4, in general, While this paper highlights the importance of binders and the solvent-free manufacturing process, it would benefit from elaboration on the broader context and the implications for LIBs'  (the battery itself) performance or energy storage, safety, and environmental impact (sustainability).

- Other: Some abbreviations, for example 3C, are not explained. Check for all abbreviations. Also, check the MDPI formatting requirement after Section X.X.X.

Author Response

Responds to the reviewer’s comments:

Reviewer 2:

Comments: This is a review paper for a solvent-free procedure for lithium-ion batteries (LIBs). The article discusses the role of binders, mainly focusing on Polytetrafluoroethylene (PTFE), and highlights the transition from traditional slurry-casting (SC) procedures to a more cost-effective and solvent-free (SF) manufacturing method. The overarching goal is to inspire future research aimed at enhancing the quality of and advancing the evolution of the SF manufacturing process for LIBs. Below are some improvements to consider.

Answer: Thank you for your positive comments.

Comments (1): Introduction. Please add a short discussion of why the current practices of manufacturing LIBs are not ecologically friendly. Add some statistics if possible.

Answer: Thank you very much for your suggestion. We have reworked the introduction section with the content on the energy required for NMP recycling and battery manufacturing depletion in lines 40-57. In addition, we have re-discussed the SF process to improve sustainable production in lines 62-80 and section 4.1.

Comments (2): Also, provide the breakdown of the review paper. Line 76-79. In a new paragraph, please expand on how the review paper followed the PRISMA guidelines. What databases were used, the keywords, and how many papers were reviewed? Refer to Systematic Review: https://www.mdpi.com/about/article_types

Answer: Thank you very much for your question. We have re-quoted and rewritten this section. The rewritten portion can be found in lines 94 to 115 in the manuscript.

Comments (3): Lack of Methodology discussion: The review paper does not provide general details about the methodology used, the differences in the methodology process, or specific results obtained between polymer fibrillation and vapor deposition. Readers may be interested in understanding how the dry binder was developed and its performance characteristics. Suggest adding another section providing a brief overview of the methodology and key to give readers a better understanding of the paper's contributions.

Answer: Thank you for your comments. We have added section 2.1, which details the development process of the dry procedure and its properties. These cover the details and differences between the six approaches.

Comments (4): Section 3.2.2. Line 317. Please briefly describe before the bullet points to match with other sections. Similarly, Section 4. Line 394.

Answer: Thank you very much for your suggestion. We have added a brief description in the corresponding section, which are detailed on lines 445, 530, and 579. This provides a more consistent structure across the various chapters.

Comments (5): Section 4.1 can be expanded rather than having seven bullet points. For example, why is the performance of semi-crystalline or non-crystalline solvent-free is undesirable (Point 3). Why preparing and storing high-molecular weight is complex? How complex? (Point 4). Why is high-quality, efficient quantitative production of electrodes tricky? What makes them tricky?

Answer: Thank you for your comments. We have analyzed and expanded on the individual points, which are now shown in chapter 5, visible in lines 534 to 633 of the manuscript.

Comments (6): Section 4.2 suggests linking the recommendation to the challenges #. Section 4, in general, while this paper highlights the importance of binders and the solvent-free manufacturing process, it would benefit from elaboration on the broader context and the implications for LIBs' (the battery itself) performance or energy storage, safety, and environmental impact (sustainability).

Answer: Thank you very much for your question. We have connected the recommendations to the challenges, which can be found in Section 5. We have also discussed the implications of the technology for the broader context, such as the impact of the SF process on LIB performance or the environment. These can be found in lines 62 to 80 in the draft, and in chapter 4.1.

Comments (7): Other: Some abbreviations, for example 3C, are not explained. Check for all abbreviations. Also, check the MDPI formatting requirement after Section X.X.X.

Answer: Thank you for your comments. We have rechecked all the abbreviations in the manuscript.

Round 2

Reviewer 1 Report

Comments and Suggestions for Authors

The authors have addressed all the comments raised by the reviewers.

Comments on the Quality of English Language

Quality of English language seems to be fine